# An Entropic Approach for Pair Trading in PSX

**DOI:** 10.3390/e25030494

**Published:** 2023-03-13

**Authors:** Laiba Amer, Tanweer Ul Islam

**Affiliations:** Department of Economics, National University of Sciences & Technology, Islamabad 44000, Pakistan

**Keywords:** pair trading, model uncertainty, model risk, optimal boundary, PSX

## Abstract

The perception in pair trading is to recognize that when two stocks move together, their prices will converge to a mean value in the future. However, finding the mean-reverted point at which the value of the pair will converge as well as the optimal boundaries of the trade is not easy, as uncertainty and model misspecifications may lead to losses. To cater to these problems, this study employed a novel entropic approach that utilizes entropy as a penalty function for the misspecification of the model. The use of entropy as a measure of risk in pair trading is a nascent idea, and this study utilized daily data for 64 companies listed on the PSX for the years 2017, 2018, and 2019 to compute their returns based on the entropic approach. The returns to these stocks were then evaluated and compared with the buy and hold strategy. The results show positive and significant returns from pair trading using an entropic approach. The entropic approach seems to have an edge to buy and hold, distance-based, and machine learning approaches in the context of the Pakistani market.

## 1. Introduction

According to quantitative models, pair trading involves a driving mechanism for mean reversions using a statistical arbitrage strategy. The perception is to recognize that when two stocks move together, their prices will converge to a mean value in the future [1]. Mean reversion allows traders to make a profit by matching a long position in one stock with an offsetting position in another stock [2]. Pair trading is an efficient method for the formation of portfolios or pair trading [3,4]; however, finding the accurate pairs and boundary points is not an easy task.

The profitability of pair trading decreased due to an increasing share of non-converging pairs [5]. To resolve the issue of non-converging pairs, several researchers contributed to the literature [6,7,8,9] and proposed cointegration as the most efficient solution for structuring pair trading [10].

After settling on how to find accurate pairs, the problem arose of how to find the mean-reverted point between them and how to identify the boundaries for when exactly the investors can buy or sell any asset. Yoshikawa [11] derived the entropy-based optimal boundary points for pair trading using Tokyo Stock Exchange 2015 data. The proposed approach for the optimal stopping problem is motivated by the work of Ekström et al. [12] and Suzuki [13]. This method is based on maximizing profit via pair trading and minimizing the relative entropy (risk). This is a robust method, as it directly tackles model misspecification [14] and provides a more persuasive solution. The choice of pairs is made through cointegration, the most effective way to identify stocks that move together [15]. Entropy has a wide application in finance as well [16,17,18].

In the context of Pakistan, there was a handful of studies conducted on pair trading [19,20], and interestingly, no one has yet considered the optimal stopping problem using stocks listed on the Pakistan Stock Exchange (PSX). This study employs the novel entropic approach proposed by Yoshikawa [11] to explore the optimal boundary points that yield the maximum profit for 64 companies listed on PSX for the period 2017–2019. The concept of maximizing the profit in pair trading based on relative entropy is a nascent idea in the literature, and this study is the first attempt in the context of Pakistan. The performance of this entropic approach is compared with the buy and hold strategy in terms of returns.

## 2. Data & Methodology

As mentioned in the last section, this study utilized the daily data for 64 companies listed on PSX for the years 2017, 2018, and 2019. These companies cover the major sectors, including cement, chemical, automobile assembler, food and personal care products, oil and gas marketing companies, oil and gas exploration companies, power generation and distribution, refinery, and pharmaceuticals. The firms’ selection criterion was based on year-wise price earnings ratios (PER); a firm with a PER lower than the sample median value was selected in the sample. The underlying idea is that the stock below the median PER is undervalued and signifies potential for higher returns [21,22]. The choice of pairs was made through Johansen cointegration, which is the most effective way to identify stocks that move together [15]. In each year, we formulated all pairs $\left(\left(n^{2}-n\right)/2\right)$ of the selected stocks and assessed each pair for cointegration.

Keeping in view the potential jumps/structural breaks in high-frequency financial data [23], the following breakpoint unit root test proposed by Bai and Perron [24] was employed.
(1)$$\Delta y_{t}=\alpha_{0}+\alpha_{1}t+\delta y_{t-1}+\overset{p}{\underset{i=1}{\sum }}\beta_{i}\Delta y_{t-i}+\mu_{t}$$
where $\mu_{t}$ is white noise.

## 3. Ornstein–Uhlenbeck (OU) Process

Pair trading utilizes the mean reversion of the composite process of two stocks. Following Yoshikawa [11], we considered the Ornstein-Uhlenbeck (OU) process $X_{t}$ such that
(2)$$dX_{t}=-\mu \left(X_{t}-\alpha \right)dt+\sigma dB_{t},\;X_{0}=\alpha$$
where μ and σ are the positive constants, α is the mean-reversion point, and $B_{t}$ is the *p*-Brownian motion. Let $X_{t}-\alpha ={\overset{ˇ}{X}}_{t}$. Then, Equation (2) implies:(3)$$d{\overset{ˇ}{X}}_{t}=-\mu {\overset{ˇ}{X}}_{t}dt+\sigma dB_{t},\;{\overset{ˇ}{X}}_{0}=0$$

The optimal stopping problem at time *t* for the process ${\overset{ˇ}{X}}_{t}$ is defined as follows:(4)v0(t, x)=ExˇSτϵℑSUP [e−ρ(τ−t)Xˇτ]
where ℑ is the set of all stopping points of B, and ρ is the discount rate. The solution of Equation (4) gives us the trading strategy: we short pair *X* when it attains the highest value and liquidate it when *X* attains zero value. These values are specified by the above equation. Alternatively, we take the long position for *X* for zero value and liquidate it for the highest value. The superscript *S* in Equation (4) is the solution to the following:(5)Infsϵℑ {E xˇS[e−ρ(τ−t)Xˇτ]+λe−ρ(τ−t)Hxˇ [S|P]}
where λ is a positive constant and *H*(.) is a relative entropy defined as follows:(6)$$H_{\overset{ˇ}{x}}=\{\begin{array}{c} E_{\overset{ˇ}{x}}^{S}[\ln\left(\frac{dS}{dP}\right),\;S\epsilon ℑ\\ \infty ,\;otherwise \end{array}$$

Thus, the optimal boundary *b*(*t*) for Equation (4) is given as:(7)$$\ln\left(b\left(t\right)\right)+\frac{1}{\sigma^{2}}\frac{\rho }{\rho -\mu }{\left(g\left(t\right)-b\left(t\right)\right)}^{2}=\ln\left(b^{*}\right)+\frac{1}{\sigma^{2}}\frac{\rho }{\rho -\mu }{\left(b^{*}\right)}^{2}$$
where $g\left(t\right)=-\frac{\sigma^{2}}{\lambda }te^{-\mu t}\;\&\;b\left(0\right)=b^{*}$. Any investor holding pair *X* should liquidate when X touches *b*(*t*) and, if not holding *X*, should short their position when *X* touches *b*(*t*) and liquidate it when it reverts to mean zero.

## 4. Results and Discussion

From the eight selected sectors, we found 64 active firms listed on PSX for the years 2017, 2018, and 2019. After applying the PER benchmark, we got 33, 34, and 40 companies, respectively. Having selected the companies, the unit root test was applied to the time series data of these stocks to find the order of integration. All the time series are integrated of order one. This led us to find the cointegrated pairs using the Johansen cointegration test at 0.05 level of significance. We found 79 = (28 + 29 + 23 = 80 − 1) unique cointegrated pairs (one pair was repeated) out of 1869 = (528 + 561 + 780) pairs of the selected stocks in the 3-year period.

Having found the pairs, we applied the maximum likelihood method to find the parameters of the Ornstein–Uhlenbeck processes, μ, α, and σ, as given in Equation (2). MATLAB R2021b was used for the coding and estimation of these parameters. However, to compute the optimal boundary points, we needed to find the parameters ρ and λ as well. The parameter ρ is the discount rate, and the parameter λ represents the level of confidence. The lower the value of λ, the lower the confidence of the agent on the reference measure as a true probability measure among the class of all probability measures and vice versa. We used ρ=0.08978, 0.1315, and 0.1440 as per the annual report of State Bank of Pakistan for 2017, 2018, and 2019, respectively, and by following Yoshikawa [11], four cases for the parameter, λ=0.001, 0.01, 0.1, and ∞ were considered. Table 1 and Table 2 present the results for only five pairs of stocks in each year involving the top listed companies (see Appendix A, Table A1, Table A2, Table A3, Table A4, Table A5, Table A6 and Table A7 for the results of other companies). After computing the values of μ, α, and σ as furnished in Table 1, we estimated the rate of returns for different values of λ for the selected companies (Table 2). On balance, pair trading yielded optimal returns for lower values of λs, which is understandable, as the parameter lambda is linked with the penalty function. All the estimated parameter values are presented in Figure 1, Figure 2 and Figure 3 and in Table 2 for their respective years. From these figures, it Is evident that the values of the mean reversion parameter differ when the stocks in the pair are selected within the sector in comparison to when the stocks are selected across the sectors.

For the real data sets, the pair trading strategy was to set the position when the pair value touches either the mean reverted point or the boundary. For example, in Figure 1 (pair: PSO and MPLF), the mean reversion point was 60.29 where we set the position, and we liquidated the position when the pair value touched the boundary *b*(*t*). If the position was set when the pair value touches the boundary, then it was liquidated when it touched the mean reversion point α. In Figure 2 (pair: PSO and BYCO), if we set our position when the pair value touched the boundary then we would liquidate at the mean reversion point, α = 9.26. The next position was set when the pair value touched either the boundary *b*(*t*) or the mean reversion point α and liquidated following the same rule.

Following this trading strategy, we estimated the rate of returns for the 80 unique pairs of the companies for the years 2017, 2018, and 2019. Gatev et al. [6] highlighted the transaction fee as an obstacle in trading. Because the transaction cost in the Pakistan Stock Exchange is 0.15 percent and we are dealing with pair trading, we discounted our return values by 0.3 percent. Table 2 provides these return values for five pairs from each year. The return values ranged from 0.2 to 25.2 percent for the year 2017, 0.4 to 19.5 percent for the year 2018, and 1.5 to 15.7 percent for the year 2019. All positive returns confirm profitable trades, which is line with the findings in the literature [1,11]. For all cointegrated pairs (Appendix A), average return values ranged from 2.9% to 18.5% which are much higher than the return values estimated in [25], which ranged from 0.1% to 1.71% using the distance-based approach for the stocks listed on the PSX during the period 2009–2016. Shaukat et al. (2021) employed the distance-based method to select the pairs and compute returns to pair trading for financial (banks) and non-financial (cement industries) sectors with a formation period of 12 months. Cement industries yielded higher returns, whereas the banks yielded lower returns. Sohail et al. [20] estimated the return on pair trading using 80 stocks from five different sectors: banking, chemicals, cement, textile, and food and care products, all of which were listed on the PSX from 2011 to 2019. Trading periods of two and one year were used for the machine learning algorithm (clustering algorithm) and distance-based methods, respectively. The study found a maximum return of 2.07 percent for the textile sector using the distance-based approach, whereas the clustering algorithm yielded a maximum return of 2.55 percent.

The distance-based approach relies on the average squared differences between the normalized prices of stocks, and principal component analysis (PCA) is used to generate the indices of the stocks that represent the weighted average prices of the stocks to be used in the machine algorithm. By construct, PCA indices resemble those generated with the cointegration technique; we found parameters α and β such that the linear combination of the two stock prices, $\alpha p_{1}+\beta p_{2}$, yielded a stationary process, whereas the weights in PCA may not yield stationary indices. Further, both the studies [20,25] did not allow cross-sector pairing that might have caused their low returns in comparison with our study. The profitability of pair trading decreases due to non-convergence of the pairs [5], and cointegration is the most efficient method to explore converging pairs [10]. Thus, the entropic approach seems to have an edge over the distance-based and machine learning approaches in the context of the Pakistani market.

Further, to evaluate our results, we contrasted our results against the buy and hold strategy with trading periods of one quarter, annually, 2 years, and 3 years (Table 3). A trading period of one year is in line with the literature [20,25]. The rate of returns for the alternative strategy is summarized in Table 3. In general, except for 2019-Q4, the top-performing stocks made a loss for this strategy, whereas Table 2 shows pair trading provided stable profits. The buy and hold strategy has a considerable risk of human error considering the pressure of all the wrong choices one can make [26]. The optimization of the boundaries backed by the Ornstein–Uhlenbeck process allowed us to incorporate all risks, improve the profitability of pair trading, and receive maximum positive returns [27]. Therefore, we suggest the pair trading strategy while taking model uncertainty into account.

## 5. Conclusions

This study employed a novel entropic approach to explore the optimal boundary points that yield maximum profit for 64 companies listed on the Pakistan Stock Exchange (PSX) for the period 2017–2019. The concept of maximizing the profit in pair trading based on relative entropy is a nascent idea in literature, and this study is the first attempt to implement it in the context of Pakistan. The performance of this entropic approach is contrasted with the buy and hold strategy in terms of returns. The following are the key findings of the study.

The values of the mean reversion parameter differ when the stocks in the pair are selected within the sector in comparison to when the stocks are selected across the sectors.On balance, optimal returns are associated with lower values of
λs; approximately, 84 percent pairs yielded optimal returns for low values of lambda (
λ=0.001 and 0.01).The return values based on entropic pair trading approached ranges from 0.2 to 25.2 percent for the year 2017, 0.4 to 19.5 percent for the year 2018, and 1.5 to 15.7 percent for the year 2019. These values are much higher than the returns estimated in
[20,25].Based on the buy and hold strategy, all the top performing stocks make a loss.The entropic approach seems to have an edge over the buy and hold, distance-based, and machine learning approaches in the context of the Pakistani market.

Pair trading is an efficient method that allows maximization of profitability by eliminating short-term price deviations in favor of long-term historical pricing relationships. The entropy-based pair trading method yielded positive returns for all the cointegrated pairs tested and confirmed their profits, which is line with the findings in literature [1,11]. According to the efficient market hypothesis (EMH), an active investor cannot be more effective than the one who buys and holds. Therefore, the returns estimated from the entropic approach were contrasted against the returns estimated through the buy and hold strategy. The buy and hold strategy yielded negative returns, except for a few cases implying losses. Consequently, we suggest the pair trading strategy while taking model uncertainty into account.

## Figures and Tables

**Figure 1 entropy-25-00494-f001:**
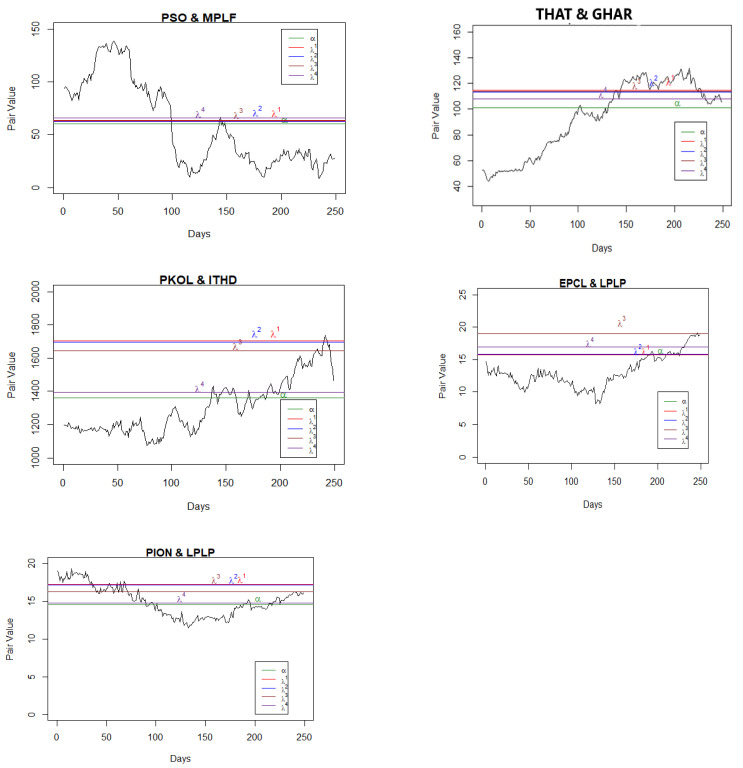
Pair values, boundaries, and mean values for the pairs (2017). Note:λ^1^, λ^2^, λ^3^, & λ^4^ are the estimated paired stock values for the given confidence levels of the agent (see Table 2).

**Figure 2 entropy-25-00494-f002:**
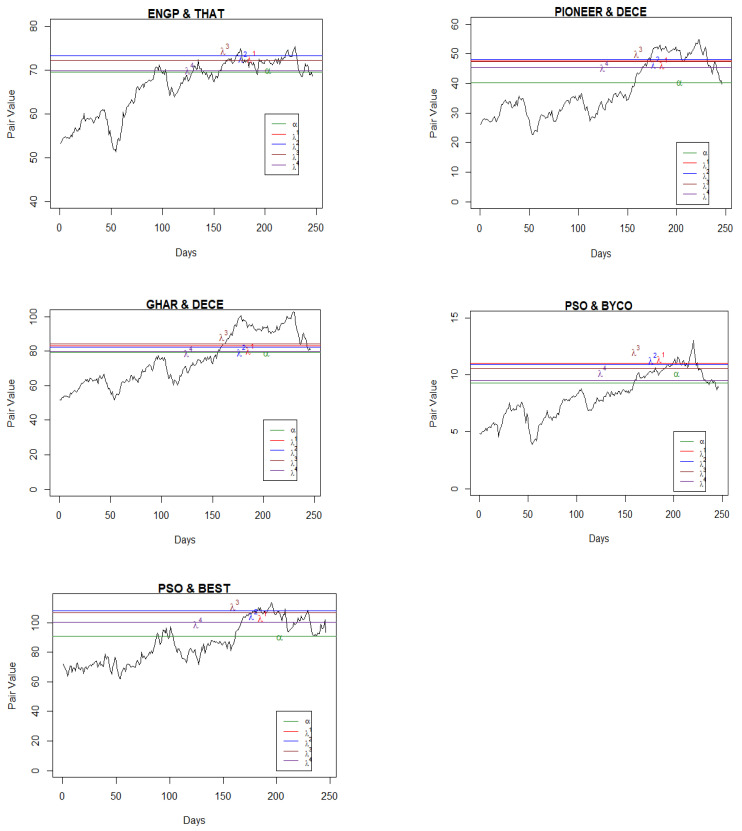
Pair values, boundaries, and mean values for the pairs (2018).

**Figure 3 entropy-25-00494-f003:**
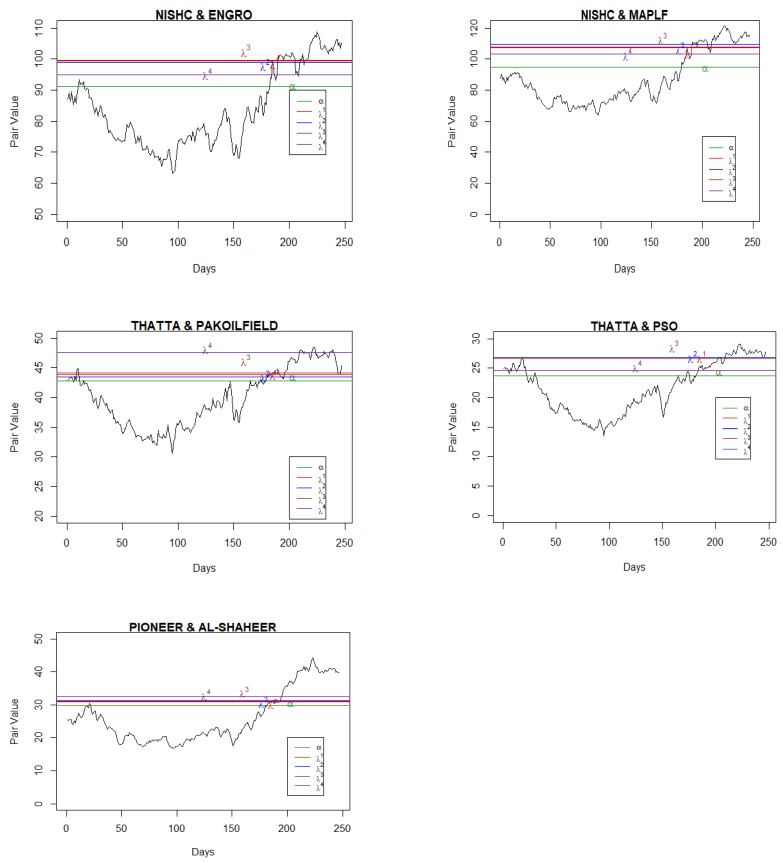
Pair values, boundaries, and mean values for the pairs (2019).

**Table 1 entropy-25-00494-t001:** Ornstein–Uhlenbeck Process parameter estimation.

Sr	Pair Name	µ	α	σ
		2017		
1	Pak State Oil (PSO) and Maple Leaf Cement LTD (MPLF)	0.04	60.29	84.91
2	Thata Cement (THAT) and Gharibwal Cement (GHAR)	0.02	101.3	39.25
3	Pak Oil Fields (PKOL) and Ittehad Chemicals LTD (ITHD)	3.74	1361.7	474.9
4	Pioneer Cement (PION) and Lalipir Power LTD (LPLP)	10.52	14.54	7.70
5	Engro Polymer and Chemical (EPCL) and Lalipir Power (LPLP)	2.29	15.71	8.47
		2018		
6	Engro Power Generation Qadirpur LTD and Thata Cement LTD	6.59	69.44	16.43
7	Gharibwal Cement LTD and Dewan Cement LTD	0.04	79.3	31.87
8	Pakistan State Oil Company LTD and Best Way Cement LTD	0.02	90.7	50.78
9	Pakistan State Oil Company LTD and Byco Petroleum Pak LTD	5.37	9.26	5.25
10	Pioneer Cement LTD and Dewan Cement LTD	0.13	40.14	19.14
		2019		
11	Nishat Chunnian Power LTD and Engro Polymer and Chemical LTD	0.01	91.05	35.11
12	Nishat Chunnian Power LTD and Maple Leaf Cement Factory	0.004	94.76	39.29
13	Thata Cement LTD and Pakistan State Oil Company LTD	1.85	23.69	10.38
14	Thata Cement LTD and Pakistan Oilfields LTD	0.01	42.82	14.44
15	Pioneer Cement LTD and Al Shaheer Corporation LTD	0.06	29.87	13.36

**Table 2 entropy-25-00494-t002:** Rate of returns for different values of *λ*.

Pair Name	*λ* = 0.001	*λ* = 0.01	*λ* = 0.1	*λ = +* *∞*
	2017			
PSO and MPLF	0.042 (62.8)	0.037 (62.5)	0.051 (63.4)	0.087 (65.6)
THAT and GHAR	0.132 (114.7)	0.12 (113.5)	0.122 (113.7)	0.065 (108.0)
PKOL and ITHD	0.252 (1704.5)	0.246 (1697.3)	0.206 (1642.5)	0.023 (1393.5)
PION and LPLP	0.186 (17.2)	0.18 (17.2)	0.114 (16.2)	0.013 (14.7)
EPCL and LPLP	0.002 (15.7)	0.006 (15.8)	0.208 (19.0)	0.08 (17.0)
	2018			
ENGP and THAT	0.055 (73.3)	0.054 (73.2)	0.04 (72.2)	0.004 (70.0)
GHAR and DECE	0.05 (83.3)	0.038 (82.2)	0.064 (84.5)	0.006 (80.0)
PSO and BEST	0.177 (106.8)	0.192 (108.2)	0.175 (106.6)	0.103 (92.6)
PSO and BYCO	0.187 (11.0)	0.177 (11.0)	0.14 (10.6)	0.022 (9.5)
PION and DECE	0.187 (47.6)	0.195 (47.9)	0.178 (47.3)	0.131 (45.4)
	2019			
NCPL and ENGRO	0.094 (99.6)	0.087 (99.0)	0.091 (99.3)	0.041 (95.0)
NCPL and MPLF	0.131 (107.2)	0.157 (109.7)	0.135 (107.5)	0.089 (103.3)
THAT and PSO	0.131 (26.8)	0.132 (26.8)	0.121 (26.6)	0.039 (24.6)
THAT and PKOIL	0.024 (43.9)	0.015 (43.4)	0.03 (44.1)	0.111 (47.8)
PION and ALSHAHEER	0.034 (30.9)	0.049 (31.3)	0.038 (31.0)	0.089 (32.5)

Estimated stock pair values for the given confidence levels of the agent are in parentheses.

**Table 3 entropy-25-00494-t003:** Rate of returns from the buy and hold strategy.

	Returns with Trading Period											
Company Name	2017				2018				2019			
	Q1	Q2	Q3	Q4	Q1	Q2	Q3	Q4	Q1	Q2	Q3	Q4
Pak State Oil	−4.7	−7.6	19.9	−20.4	8.8	−3.5	0.5	−15.1	−8.4	−20.3	−7.5	19.4
Thata Cement LTD	11.7	−10.5	−32.7	−14.1	13.7	−15.7	−14.3	−24.4	−8.7	−24.1	−24.2	56.1
Pioneer Cement LTD	0.8	−7.8	−27.4	−28.4	10.2	−35.6	−4.1	−5.2	−21.0	−33.8	−15.3	51.3
Nishat Chunnian Power	−14.8	−5.3	−10.1	−20.0	−6.9	−9.2	−8.6	−1.1	−4.7	−16.1	4.7	11.3
Gharibwal Cement LTD	17.3	−24.3	−23.3	−26.8	6.3	−22.8	−8.3	−20.0	−17.6	−21.5	−17.2	53.1
	1-Year						2-Year				3-Year	
	2017		2018		2019		2017–2018		2018–2019		2017–2019	
Pak State Oil	−20.9		−8.6		−2.4		−27.0		−6.5		−25.4	
Thata Cement LTD	−44.9		−37.8		−17.8		−66.3		90.3		−71.6	
Pioneer Cement LTD	−55.6		−33.7		−30.8		−70.4		−51.9		−78.5	
Nishat Chunnian	−43.2		−28.5		−19.0		−58.3		−42.1		−66.2	
Gharibwal Cement LTD	−53.5		−36.2		−14.0		−52.0		−44.9		−73.6	

## Data Availability

Data is available publicly at https://www.investing.com/.

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
