# Peer review of "An Entropic Approach for Pair Trading in PSX"

_entropy, 2023, doi:10.3390/e25030494_

Round 1
Reviewer 1 Report
I suggest the authors to recompose the introduction section so that the novelty of the paper can be emphasized in a more informative way.
Author Response
Entropy
Manuscript ID: entropy-2138553
Dear Reviewer,
We appreciate the time and effort you have dedicated to providing insightful feedback on ways to strengthen our paper. Thus, I am pleased to resubmit my article for further consideration. I have incorporated changes that reflect the detailed suggestions you have graciously provided. We also hope that our edits and responses below satisfactorily address all the issues and concerns you have noted.
The following is a point-by-point response to the questions and comments to facilitate your review of my revisions.
Comments & Responses:
- I suggest the authors recompose the introduction section so that the novelty of the paper can be emphasized in a more informative way.
Response: The introduction section is rewritten as per the instructions (pages #1-2). Further, a serious revision is conducted to remove all the grammar and spelling mistakes.
The conclusions are supported by the results.
Reviewer 2 Report
In this manuscript, the authors use an entropy approach to study optimal boundary points that bring maximum profit for 64 companies listed on the Pakistan Stock Exchange for the period 2017-2019. The concept of profit maximization in pair trading based on relative entropy is still a new idea in the literature. The authors propose a strategy for trading pairs taking into account the uncertainty of the model. I have no comments on the abstract, but in the introduction I would like to see the hypothesis of the study.
The sections data and methodology and results and discussions are well written, there are also no comments on these sections.
But I did not like the conclusion, since the beginning of the conclusion repeats the annotation. It is necessary to rewrite the conclusion to reflect in the conclusion the main results, conclusions on all paragraphs of the manuscript. It is necessary to write about the prospects for further research of the problems considered in the article.
After eliminating these comments, I can recommend the manuscript for publication in the journal.
Author Response
Entropy
Manuscript ID: entropy-2138553
Dear Reviewer,
We appreciate the time and effort you have dedicated to providing insightful feedback on ways to strengthen our paper. Thus, I am pleased to resubmit my article for further consideration. I have incorporated changes that reflect the detailed suggestions you have graciously provided. We also hope that our edits and responses below satisfactorily address all the issues and concerns you have noted.
The following is a point-by-point response to the questions and comments to facilitate your review of my revisions.
Comments & Responses:
- The sections on data and methodology and results and discussions are well written, there are also no comments on these sections.
Response: Thank you!
- But I did not like the conclusion, since the beginning of the conclusion repeats the annotation. It is necessary to rewrite the conclusion to reflect in the conclusion the main results, and conclusions in all paragraphs of the manuscript. It is necessary to write about the prospects for further research into the problems considered in the article.
Response: The conclusion section is revised, and the main results are included in this section. Further, a serious revision of the manuscript is conducted to remove all grammar and spelling mistakes.
Reviewer 3 Report
Please refer to my review report.

Author Response
Entropy
Manuscript ID: entropy-2138553
We appreciate the time and effort you have dedicated to providing insightful feedback on ways to strengthen our paper. Thus, I am pleased to resubmit my article for further consideration. I have incorporated changes that reflect the detailed suggestions you have graciously provided. We also hope that our edits and responses below satisfactorily address all the issues and concerns you have noted.
The following is a point-by-point response to the questions and comments to facilitate your review of my revisions.

Round 2
Reviewer 1 Report
Well done. Congrats.
Author Response
Dear Reviewer,
We appreciate the time and effort you have dedicated to providing insightful feedback on ways to strengthen our paper.
Thank you very much for accepting our all responses to your previous comments and valuable suggestions.
Reviewer 2 Report
Dear authors!
All comments have been taken into account and eliminated. The article can be accepted for publication in the journal.
Reviewer 3 Report
Please refer to my review report.

Author Response
Entropy
Manuscript ID: entropy-2138553
Dear Reviewer,
We appreciate the time and effort you have dedicated to providing insightful feedback on ways to strengthen our paper. Thus, I am pleased to resubmit my article for further consideration. I have incorporated changes that reflect the detailed suggestions you have graciously provided. We also hope that our edits and responses attached satisfactorily address all the issues and concerns you have noted.
Attached is a point-by-point response to the questions and comments to facilitate your review of my revisions.

Round 3
Reviewer 3 Report
I would appreciate that the authors made efforts addressing my comments and suggestions. I would suggest accepting the paper in its current form.